# Explainable Few-Shot Learning for Multiple Sclerosis Detection in Low-Data Regime

Anonymized

**Abstract.** Diagnosing multiple sclerosis (MS) accurately is highly challenging due to symptom overlap with other demyelinating diseases. Here, we present DemyeliNeXt, an explainable few-shot learning framework designed to classify MS and other demyelinating diseases from MRI scans. This framework employs a prototypical network with a 3D DenseNet-121 backbone and uses Deep SHAP for feature importance visualization. We train our DemyeliNeXt on a dataset from African populations and we test it for different datasets including MICCAI MSSEG2 public dataset. Our findings demonstrate robust performance across diverse datasets highlighting the model's potential to enhance diagnosis accuracy and generalizability in various clinical settings.

**Keywords:** Few-Shot Learning · Explainable AI · Multiple Sclerosis · 3D MRI · and Deep Learning

## 1 Introduction

Multiple sclerosis (MS) is a complex neurological condition that is often misdiagnosed due to its symptom overlap with other conditions such as vasculitis and vascular leukoencephalopathy. Studies indicate that over half of the patients were misdiagnosed for a period exceeding three years. Moreover, 70% of these patients had been administered disease-modifying therapies (DMTs), and 31% suffered unnecessary morbidity due to the incorrect diagnosis and treatment [2,13]. This diagnostic challenge results in a prolonged time to achieve a definitive diagnosis, often exceeding several months. Hence, accurate and timely diagnosis is crucial for effective management and treatment planning in MS patients. Advanced imaging techniques and biomarker analyses are increasingly important in differentiating MS from other similar presenting conditions, thereby reducing diagnostic errors and improving patient outcomes. Machine learning provides a robust approach for the analysis of medical images and the diagnosis of MS.

In this context, several studies have employed machine learning models for MS classification. For instance, Wang et al. [15] employed a multi-layer convolutional neural network (CNN) with data augmentation techniques to classify MS. However, the model's lack of explainability raises concerns about the potential misclassification of MS scans due to reliance on spurious correlations. To address this issue, Zhang et al. [17] proposed a classification model for MS subtypes based on VGG19 [11] with global average pooling and utilized Grad-CAM++ [1] for model explanation. While effective in performance and interpretability,

this approach did not account for the diversity of MS data, particularly by not comparing it with other similar demyelinating diseases such as RON and vasculitis. To rectify this concern, Huang et al. [3] leveraged a Transformer-based model with a Multiple Instance Learning (MIL) strategy to discriminate between MS and various demyelinating diseases. The authors used Grad-CAM to visualize feature extraction through activation heatmaps. Nevertheless, their study did not incorporate data from low-income countries, such as datasets from the African population. This omission underscores a critical gap, as regional genetic and environmental factors influence disease onset and progression [16]. These factors impact the timeliness and accuracy of MS diagnosis, thereby potentially threatening the patient's life.

In this paper, we introduce DemyeliNeXt, an explainable few-shot learning framework for the classification of MS and other demyelinating diseases. Our approach employs a prototypical network with a 3D DenseNet-121 backbone, which integrates spatial information from FLAIR (Fluid Attenuated Inversion Recovery) MR (Magnetic Resonance) images to classify them as MS vs other demyelinating diseases (NON-MS). Additionally, the framework provides model interpretability through the Deep SHAP model for visualizing the most important features leading to the classification of the input MRI. The primary contributions of our work are as follows:

1. Application of Few-Shot Learning: We apply few-shot learning for the detection of multiple sclerosis (MS).
2. Emphasis on Explainability: Our method integrates explainability mechanisms to enhance interpretability, making it more suitable for clinical settings.
3. Utilization of African 3D MRI Data: We trained our model using 3D MRI data from African populations, which are often underrepresented in medical datasets. By benchmarking our model against MICCAI MS public dataset, we demonstrated its robust performance, thereby validating its generalizability across diverse populations.

## 2    Proposed Method

In this section, we explain the key building blocks of our proposed DemyeliNeXt architecture for explainable MS identification from other demyelinating diseases. **Table** 1 displays the key mathematical notations used throughout our paper.

### 2.1    Architecture overview

In this study, we introduce DemyeliNeXt, a four-stage pipeline designed for the classification of multiple sclerosis (MS) and other demyelinating diseases from MRI scans, while also providing model interpretability. Figure 1 illustrates the first stage (Section 2.2), which involves a preprocessing pipeline for FLAIR MRI scans. Here, raw FLAIR images are normalized, while noise and artifacts are

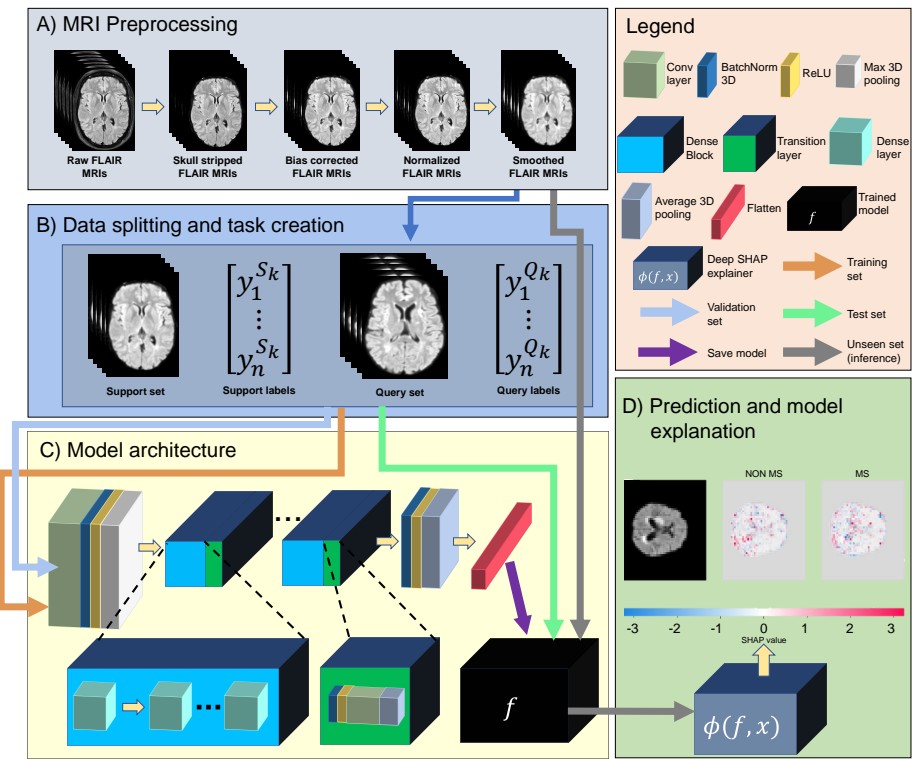

**Fig. 1.** *DemyeliNeXt Pipeline.* (A) Preprocessing MRI scans: includes skull stripping, bias correction normalization, and FLAIR MRI smoothing. (B) Data splitting into support and query sets. (C) Training a prototypical network with 3D DenseNet-121 backbone. (D) Model testing on unseen MRIs with explanations provided using Deep SHAP.

reduced. In the second stage, the MRI scans are divided into training, validation, and testing sets. Each set contains a support set ($S$) with labeled examples to update model parameters and a query set ($Q$) with unlabeled examples for performance evaluation.

The third stage (Section 2.3) involves training a 3D DenseNet-based (DenseNet-121) [4] prototypical network to classify the preprocessed MRIs. The training process utilizes $N^{tr}$ training tasks, each comprising $N_{shots}$ support examples for model weight updates and $N_{query}$ query examples for performance assessment. In the final stage, we employ Deep SHAP [8] to approximate the model for interpretability. Deep SHAP, inspired by DeepLIFT [10], assigns importance scores to each input feature by propagating neuron contributions backward through the network. These scores are based on the difference from a reference input, known as the "baseline" or "background" input, representing a typical or neutral state for the input features. The importance scores are computed via the combination

of the model's weights, the actual input and the baseline input. After training the explainer, we use the model and explainer to predict and interpret new examples of MS and other demyelinating diseases during inference.

**Table 1.** Major mathematical notations

| Mathematical notation | Definition |
|:---:|:---:|
| $K$ | Number of classes |
| $S$ | Support set |
| $S_k$ | Support set labeled with class $k$ |
| $Q$ | Query set |
| $Q_k$ | Query set labeled with class $k$ |
| $N$ | Number of labeled examples |
| $N^{tr}$ | Number of training tasks/episodes |
| $N^{val}$ | Number of validation tasks/episodes |
| $N^{ts}$ | Number of testing tasks/episodes |
| $N_{shots}$ | Number of learning shots |
| $N_{query}$ | Number of query examples |
| $x$ | Input image |
| $f$ | Deep learning model |
| $\phi(f, y)$ | Explanation model |

### 2.2 Preprocessing Pipeline

We begin our preprocessing pipeline by anonymizing DICOM MRI scans, converting them to NIfTI format. This process removes patient metadata and consolidates each volume into a single file. Next, we perform skull stripping using the ROBEX algorithm [5] to eliminate non-brain tissues. We then apply bias field correction using the N4ITK algorithm [14] to remove low-frequency intensity non-uniformities. Following this, we normalize MRI intensities to a range of 0 to 1. We reduce the noise using a Gaussian filter. Finally, we reorient the images to the "IPL" (Inferior, Posterior, Left) orientation, resample them to isotropic voxels, and resize them to a standard format.

### 2.3 Few shot learning

**Prototypical network.** Prototypical Networks [12] seek to find a metric space in which samples from the same class are close to one another. This approach makes the model particularly useful in settings with limited labeled data. Based on the prototype concept [12], the model depicts each class using the mean of its embedded support set $S$. Prototypical Networks then determine query samples $Q$ based on their proximity to these prototypes. To generate the image embeddings, we use a 3D DenseNet-121 [4] as a backbone. We create dataset tasks using a sampler that follows uniform distribution to load data from the dataset for each label.

**Loss function** We use binary cross-entropy loss:

$$\mathcal{L} = - \left[ y \log(p) + (1 - y) \log(1 - p) \right] \tag{1}$$

where $y$ and $p$ are the MS label and the predicted probability of MS from the model respectively. We use ADAM [7] as an optimizer with step LR scheduler to decay the learning rate.

### 2.4   Explainability with Deep SHAP

Deep SHAP [8] approximates explanations for deep neural network models using SHAP (SHapley Additive exPlanations) values to quantify feature importance. This method integrates concepts from a deep learning explanation technique called DeepLIFT [10] that uses Shapley values [9]. We apply Deep SHAP to interpret our trained 3D DenseNet-based ProtoNet model using preprocessed MRI scans from the testing dataset. This approach creates a simplified explanation model, assessing the importance of each voxel in our testing MRIs, visualized through feature importance plots.

### 2.5   Model inference and explanation

After training and evaluating the model, we perform inference on unseen examples where we pass them to the explainer to check the used feature importance of the model on the classification of the new examples.

## 3   Results and discussion

In this section, we provide a quantitative evaluation of our model on three distinct datasets and we display the findings of the used Deep SHAP.

### 3.1   Evaluation datasets

In this work, we utilized three datasets, summarized in Table 2. We trained, validated, and tested using a set that comprises 184 FLAIR MRI scans from 110 patients with multiple sclerosis (MS) and other demyelinating diseases (NON-MS) which we split into three different sets as follows: 70% for training, 15% for validation and 15% for testing. This dataset is sourced from the radiology department at CHU X (disclosed upon acceptance). It includes 92 3D and axial scans: 56 from MS and 36 from other demyelinating diseases such as vasculitis and vascular leukopathy.

We tested our model on a set containing 91 FLAIR MRI scans from 34 MS patients, obtained from the MRI center of CHU Y (anonymized). Additionally, we used 80 3D FLAIR MRI scans from 40 patients in the MICCAI 2021 MS Segmentation Challenge (MSSEG-2) as a benchmark dataset. We randomly sampled data from each set to create tasks consisting of a support set and a query set. Prior to training, gamma correction was applied to all scans using $\gamma = 2.5$. No further data augmentation was performed.

**Table 2.** Datasets statistics

| Source | Number of patients | Number of scans | Age | Gender |
|--------|--------------------|-----------------|------|--------|
| CHU X(anonymized) | 54 MS / 56 NON-MS | 92 | 21-63 | 21M / 32F |
| CHU Y(anonymized) | 34 MS | 91 | NA | 4M / 31F |
| MSSEG-2 | 40 MS | 80 | NA | NA |

### 3.2 Experimental settings.

**Parameter settings** For model training, we used an ADAM optimizer [6] with a learning rate of 0.001. We applied learning rate decay for every single step by 0.1 using a step scheduler. As for Deep SHAP explainer training, we adopted 90 background examples. We trained our model and our explainer on the Nvidia RTX 3090 GPU.

**Hyperparameter Settings** We conducted three distinct training experiments using 2-way ($K = 2$) classification. Validation was performed with 100 tasks ($N^{val} = 100$) every 500 training tasks. Testing was also conducted with 100 tasks. Each training lasted for 1000 episodes. Detailed hyperparameters for each experiment are listed below:

- **Experiment A**: Trained with 5 examples in both support and query sets ($N_{shots} = 5$, $N_{query} = 5$) for 1000 episodes.
- **Experiment B**: Trained with 3 examples in both support and query sets ($N_{shots} = 3$, $N_{query} = 3$).
- **Experiment C**: Trained with 1 example in both support and query sets ($N_{shots} = 1$, $N_{query} = 1$).
- **Experiment D**: We used the saved model from Experiment A, to test on datasets from CHU Y MS and CHU X NON-MS.
- **Experiment E**: We used the saved model from Experiment A to test on datasets from MSSEG-2 and CHU X NON-MS.

**Table 3.** Experiments results

| Experiments | Accuracy | Precision | Recall | Specificity | F1-score |
|-------------|----------|-----------|--------|-------------|----------|
| A: 5 shots 5 queries (Dataset: CHU X) | 72.2% | 0.77 | 0.63 | 0.82 | 0.69 |
| B: 3 shots 3 queries (Dataset: CHU X) | 76.17% | 0.75 | 0.79 | 0.73 | 0.77 |
| C: 1 shot 1 query (Dataset: CHU X) | 61.5% | 0.63 | 0.57 | 0.66 | 0.6 |
| D: 5 shots 5 queries (Dataset: CHU Y MS + CHU X NON-MS) | 75.4% | 0.75 | 0.75 | 0.76 | 0.76 |
| E: 5 shots 5 queries (Dataset: MSSEG-2 + CHU X NON MS) | **98.2%** | **0.98** | **0.98** | **0.98** | **0.98** |

### 3.3   DemyeliNeXt evaluation

Table 3 shows the classification accuracy, precision, recall, specificity, and F1 scores for the different experiments detailed in Section 3.2. Across all experiments, Experiment E, which involved training on an African dataset and testing on a combination of African and European datasets, achieved the highest classification accuracy. This result may indicate that our model has the ability to generalize well across different populations despite the differences in socio-economic conditions between the subjects in each of the datasets.

In contrast, Experiment C, which utilized only one shot and one query, demonstrated the lowest performance. This indicates that reducing the number of shots below a certain threshold adversely affects model accuracy. These findings suggest that while reducing shots can decrease computational demands, maintaining an adequate number of shots is critical for reliable performance (see experiments A and B). In particular, one could generally recommend using the models trained in Experiments A and B as a guide for practitioners in balancing computational efficiency with diagnosis accuracy for MS.

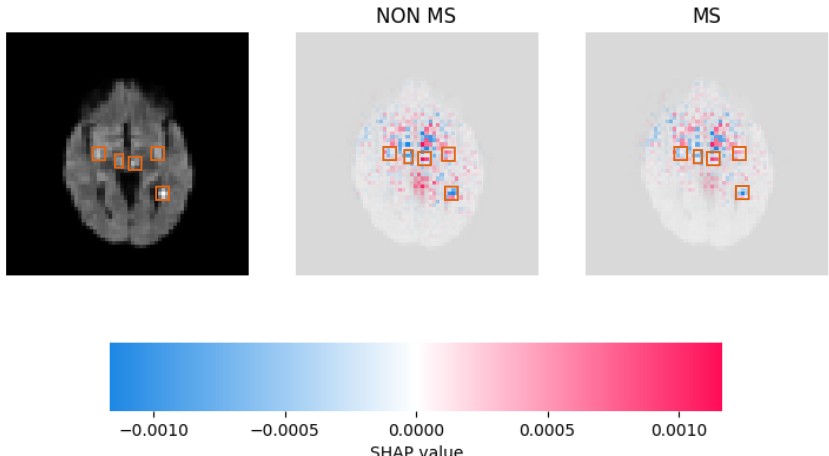

**Fig. 2.**  *Deep SHAP Explanation for NON-MS Example.* The left panel displays an annotated MRI section of a patient with a NON-MS demyelinating disease. The center panel highlights the features identified by our model for classifying the case as NON-MS using Deep SHAP. The right panel shows the features identified for classification as MS using Deep SHAP. Lesions' locations are highlighted with orange rectangles across all panels. For the two right hand side panels, blue indicates the features excluded by the model, while red shows the important features for each class

Figure 2 illustrates the explanation of our model on an unseen NON-MS example with lesion annotation. The plot highlights the features utilized by our trained ProtoNet model for classification that are explained by the Deep SHAP

method. The Deep SHAP explainer seems to identify some of the key features for classification. However, it also included features that are deemed irrelevant to clinicians. This could indicate that while the model is effective in feature identification, there is a need for further refinement to align its outputs with clinical relevance.

**Limitations and future studies.** Despite the promising results, DemyeliNeXt has a few limitations that warrant further investigation. For instance, our approach currently utilizes only FLAIR MRI scans; incorporating other imaging modalities like T1-weighted and T2-weighted MRIs could potentially enhance diagnostic accuracy. While Deep SHAP provides some level of explainability, the clinical relevance of the highlighted features remains uncertain, indicating a need for further refinement. In future studies we aim to focus on expanding the dataset to include diverse minority populations, integrating multimodal imaging techniques, as well as developing more clinically relevant explainability methods.

## 4    Conclusion

In this study, we introduced DemyeliNeXt, an explainable few-shot learning framework designed for the classification of multiple sclerosis (MS) and other demyelinating diseases in an African population. By incorporating the Deep SHAP model, we provided visual explanations for the model's decisions, enhancing its interpretability. Our findings, derived from MRI data of underrepresented African populations, demonstrate that this approach can generalize effectively to non-African datasets. Although the classification accuracy decreases with fewer shots, the method remains computationally efficient and can aid practitioners in improving diagnostic accuracy. In future work, we aim to extend our framework by including more minority populations and integrating additional neuroimaging modalities, thereby enhancing the generalizability and robustness of our model.

**Code availability.** We provide the code repository of our method on GitHub at this link: https://github.com/.../

**Acknowledgments.** Disclosed upon acceptance

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
