# OpenReview forum: "Explainable Few-Shot Learning for Multiple Sclerosis Detection in Low-Data Regime"
_MICCAI.org/2024/Workshop/MSB — MICCAI Student Board EMERGE Workshop 2024 Oral_

### Official Review · Reviewer_nxmY · 2024-07-03

**Recommendation:** 2
**Confidence:** 4

**Clarity:**

The paper is generally clear but has some clarity issues that could be addressed with moderate revision

**Feedback:**

Besides addressing all points listed in the "Weaknesses" section, the following considerations might also be helpful:

- Section 3.1 is confusing and should be improved. This is the most important part where the reader can understand how the evaluation is performed. Additionally, only a simple train-validation-test split is performed. More convincing results might be obtained by performing a cross-validation analysis (possibly using "clogging"(*) although not absolutely required).
- The schematic representation proposed in Fig 1 should be improved. The main purpose of having a diagram is to help the reader understand the proposed pipeline. Currently, this is not the case since the diagram increases the confusion on how the training and testing are actually performed.
- Without a fair comparison with the literature, it is not possible to evaluate the proposed approach. Additionally, this work seems more of an application of an existing approach rather than a completely new method as confirmed by the authors in Section 1. In fact, changing a few layers of a classification network does not make it a new method. Thus, in my opinion, giving a new name to an existing method is only confusing.


(*) D. K. Barrow and S. F. Crone, "Crogging (cross-validation aggregation) for forecasting — A novel algorithm of neural network ensembles on time series subsamples," The 2013 International Joint Conference on Neural Networks (IJCNN), Dallas, TX, USA, 2013, pp. 1-8, doi: 10.1109/IJCNN.2013.6706740.

**Justification:**

Although potentially a very interesting work, the main concern is related to data leakage from the evaluation strategy detailed in Section 3.1. Additionally, it is not clear how the authors addressed the different resolutions between various datasets, which may (and probably is) the main reason for the good results reported in Table 3 (especially for MSSEG-2).

**Reproducibility:**

Not enough amount of details available for reproducing the main results, and open access details are unclear

**Strengths:**

- The use of few-shot learning, which represents an important area of research in medical image analysis due to the limited amount of labeled data.
- The use of African populations MS data since regional, genetic, and environmental factors influence disease onset and progression.
- Explainability using Deep SHAP model
- Benchmarking on the MICCAI MS public dataset: MSSEG-2
- Ablations using multiple combinations of N_shots and N_query

**Summary:**

This work introduces the DemyeliNeXt model, which aims to use few-shot learning for the classification of MS and other demyelinating diseases. An additional interpretability model (Deep SHAP) was also used to visualize the most important features leading to the classification.

**Weaknesses:**

- The work limits the analysis to a binary classification task. It might be more interesting to perform a multiclass classification task to investigate the ability of the model to detect MS and improve interpretability.
- Figure 1 does not help understanding the pipeline. For instance, the arrows from box B to box C are unclear. It seems that both the training and validation are used as input to the model and that the query from the test set is fed inside a new black box model "f", although I am not sure how to interpret it.
- The strategy used to validate the model is wrong. Since we are dealing with repeated samples in the dataset, splitting the data in training, validation and testing leads to data leakage since scans from the same patients might be in both training-validation set and test set invalidating the results.
- Section 3.1 Evaluation datasets should be renamed. It describes the data used in both train-validation as well as testing.
- The description of the dataset in Section 3.1 is unclear.
   - "184 FLAIR MRI scans from 110 patients with multiple sclerosis (MS) and other demyelinating diseases (NON- MS)" are used. What are
       these "other demyelinating diseases"? Probably vasculitis and vascular leukopathy although the exact distribution should be given.
   - "This dataset is sourced from the radiology department at CHU X (disclosed upon acceptance). It includes 92 3D and axial scans".
       What are these 92 scans? Weren't they 184 FLAIR MRIs?
- It is not clear how the support set and query set are used in testing phase. Is the support set labeled? Further clarifications are needed to understand how the dataset is used for obtaining results in Tab. 3 which at the moment is not clear.
- The quality of the image presented in Fig 2 is quite low. It seems that the high performances obtained when comparing with MSSEG-2 data are mostly due to the differences in image resolution (and the leakage from the splitting strategy) rather than the model performance
- No comparison with other methods proposed in the literature is provided.
- Link to the code on the GitHub repository is not working at the time in which I am reviewing this paper, although this might be due to the anonymity constraint.

---

> ### Author Response · Authors · 2024-07-14
> **Rebuttal by Authors**
>
> - We thank the reviewer their valuable remark, and we apologize for this shortcoming. While we acknowledge that performing multi-class classification would be more interesting, unfortunately not all the provided data is labeled with the respective diseases. Rather the dataset is annotated by either MS or non MS demylinating diseases.  We added the following sentence in the introduction to clarify this more. ''This study focuses on distinguishing MS from other demyelinating diseases."
> - We thank the reviewer for their feedback about Figure 1. We have updated our figure for more clarification of our pipeline.
> - We thank the reviewer for pointing this out and we apologize for lack of clarity. We rerun the same experiments using random patient-wise split to avoid data leakage as mentioned. Also, we updated our results in the table 3 as well as the split description: "The dataset was split randomly and patient-wise into three different sets as follows..."
> - We thank the reviewer for the feedback concerning the section renaming. We apologize for this shortcoming. We have changed the title to "Employed datasets".
> - We thank the reviewer for the feedback concerning the confusion regarding the first dataset description. We apologize for the lack of clarity. We have reformulated the text to be clear and concise in order to state the dataset details in terms of size for each class from each source. The first dataset is sourced from CHU X. The total size of this dataset is 182 scans distributed equally between MS and non-MS: 91 scans from 51 MS patients and 91 scans from 67 patients with other demyelinating diseases. ''We trained,
> validated, and tested using a set that comprises 182 FLAIR MRI scans from 109 patients with multiple sclerosis (MS) and other demyelinating diseases (NON-MS). ... It includes 3D and axial scans: 91 scans from 51 MS patients and 91 scans from 67 patients with other demyelinating diseases..."
> - We thank the reviewer for their feedback concerning meta-testing and experimental results. We apologize for the lack of clarity.
> In few-shot learning, we either test on unseen examples or unseen categories. The meta-testing set contains support and query sets like training to calculate the distance between them like in meta-training phase. We have reformulated the text to show that only one dataset used for training the model, which is from CHU X. We split the dataset into train, validation and test sets. After training we saved the model. After that, we evaluated the trained model further on 2 MS datasets: 1 from CHU Y and the other is from MSSEG-2 dataset. To address the issue of the class imbalance, we used NON-MS samples from the testing set of CHU X in both experiments. We hope that the the answer is clear for the reviewer
> - We thank the reviewer for their feedback. We have retrained our model with a careful train, validation and test splitting to avoid any potential data leakage. The results are consistent with the previous versions. For the picture resolution, it is affected by the image resizing. All scans are resized to the same resolution before used for training, validation, testing and explaining. as mentioned in the paper in Section 2.2 "resize them to a standard form"
> - We thank the reviewer for the feedback. Unfortunately the submission deadline prevented us from performing extra experiments.
> - We thank the reviewer for their feedback concerning the code availability. We have put our code in an anonymous links as follows: https://we.tl/t-WNxF1OZ8nj. We will reveal the GitHub repository link upon acceptance.
> - We thank the reviewer for the feedback concerning the usage cross-validation. Unfortunately, the timeline for this study was tight, we didn't have the time to execute al the experiments at this stage. In future studies, we aim to perform a more detailed analysis of our method.
> - We thank the reviewer for their feedback. Regarding more experiments: we aim to perform more experiment in an extended version of this study. Regrading naming the model, we chose to give it a name for ease of reference and finding our study in the future.

---

### Official Review · Reviewer_Z9ws · 2024-07-04

**Recommendation:** 5
**Confidence:** 5

**Clarity:**

The paper is generally clear but has some clarity issues that could be addressed with moderate revision

**Feedback:**

"Studies indicate that over half of the patients were misdiagnosed for a period exceeding three years." Please provide reference for this statement.

Please explain why only the FLAIR sequence of MRI was used, compared to other sequences and the combination of all MRI sequences.

In figure 1, it is difficult to read panel C by referring back and forth between panel C and the legend. Since the symbols in the legend appear only once or twice in panel C, it would be more reader friendly by removing the legend panel, and enlarging panel C and putting the legend text besides the symbols. This can save some space by removing redundant symbols in the legend and make the figure more readable.

"there is a need for further refinement to align its outputs with clinical relevance."
Please provide rationale on why refining the explanation for clinical relevant features are needed, especially considering this scenario: what if the explanation can provide clinical relevant features to explain its decision, but the decision is incorrect.
If the authors aim to improve the clinical relevance of the explanation, the clinical relevant is related to the form of explanation, rather than the specific explanation content, ref: Guidelines and evaluation of clinical explainable AI in medical image analysis https://doi.org/10.1016/j.media.2022.102684

If possible, it would be interesting to add more quantitative or qualitative results on how well the prototype features are learned.

**Justification:**

This paper contributes to the application of MS detection in underrepresented dataset, which can contribute to the MICCAI this year and the workshop.

**Reproducibility:**

Sufficient amount of details available for reproducing the main results, and open access is provided (or promised upon acceptance) to source code and/or data

**Strengths:**

1. The study topic is important to study MS in low-income countries, addressing unique challenges such as lack of health resources and data.
2. The paper is well-written with clear structures and technical details, and it is easy to read and to follow.
3. The evaluation is on model performance is relatively comprehensive.
4. The paper exhibits good research practice that lists the limitations of the proposed technique.
5. The paper shows good reproducibility by making the code available.

**Summary:**

This work proposes DemyeliNeXt that aims to address the few-shot learning and explanability problems on MS detection using MRI, with the consideration of modeling disease for low-resource regions.

**Weaknesses:**

1. It should be noted that explainable techniques require evaluation. When using an ML model on a task, it is necessary to report the model task performance. Similarly, when using an explainability technique, it is necessary to report the explainability performance. In the work, the application of the explanation technique lacks evaluation on how faithful the explanation can reflect the true model decision process. If evaluation of explanation cannot be provided in the current version, please list the lack of evaluation of explanation as a limitation, and avoid overclaims that are not backed up by evidence, such as "enhancing its interpretability".

2.  The evaluation lacks a baseline study on the comparison of the applied ProtoNet with vanilla network architecture.

---

> ### Author Response · Authors · 2024-07-14
> **Rebuttal by Authors**
>
> - We thank the reviewer for the feedback concerning the evaluation of the explanation. Unfortunately, the timeline for this rebuttal does not allow for such analysis. We updated the paper to mention that in the future works: "...as well as developing more clinically relevant explainability methods with their evaluation."
> - We thank the reviewer the feedback concerning running the benchmark experiments. Unfortunately, due to time and resource shortage, this was not possible. We mentioned that in the updated limitation and future studies subsection.
> - We thank the reviewer for their feedback concerning the references for the studies mentioned. We apologize for the shortcoming. We updated the paper with the references (Gaitán, M.I., Correale, J.: Multiple sclerosis misdiagnosis: a persistent problem to
> solve. Frontiers in Neurology 10, 466 (2019): https://www.frontiersin.org/journals/neurology/articles/10.3389/fneur.2019.00466/full and Solomon, A.J., Bourdette, D.N., Cross, A.H., Applebee, A., Skidd, P.M., Howard, D.B., Spain, R.I., Cameron, M.H., Kim, E., Mass, M.K., et al.: The contemporary spectrum of multiple sclerosis misdiagnosis: a multicenter study. Neurology 87(13), 1393–1399 (2016): https://pubmed.ncbi.nlm.nih.gov/27581217/)
> - We thank the reviewer for their feedback concerning non clarity about the choice of FLAIR sequence of MRI. We apologize for the inconvenience. The other modalities are also used in the exam. However, FLAIR sequence is used by clinicians to confirm MS diagnosis thanks to the hypertensities in the image representing the lesions. Leveraging the combination of all MRI sequences is interesting to explore. However, it is computationally expensive. In this study, we aim to make automatic MS detection affordable even in low-settings. We updated the paper to mention the reason behind FLAIR sequence in the introduction: "Furthermore, a key finding in MS identification is the presence of white matter lesions in the brain, detectable via FLAIR sequence of MRI."
> - We thank the reviewer for the feedback concerning figure 1. We have updated figure 1 accordingly.
> - We thank the reviewer for your feedback concerning clinical relevance We are sorry for the ambiguity. MS diagnosis is difficult, specially at early stage. It would be helpful if the explanation matches clinical relevant features to apply it in real world scenarios. On the other hand, we have updated the paper to warn about the explanation features: "However, one should note that there is a risk that the included features in the explanation could be deemed irrelevant to clinicians."
> - We thank the reviewer the feedback concerning adding more quantitative and qualitative benchmarks. Unfortunately, due to time and resource shortage, this was not possible.

---

### Official Review · Reviewer_wPN9 · 2024-07-05

**Recommendation:** 4
**Confidence:** 4

**Clarity:**

The paper is clear and well-written, with minor areas for improvement in clarity

**Feedback:**

- In the Introduction, it is stated "Studies indicate that over half...", I would add a reference to these studies.
- Later in the same paragraph it is mentioned "... in differentiating MS from other similar presenting conditions...". I would add the names of these conditions.
- In Figure 1, the deep blue arrow is missing from the legend.
- Table 1 defines notation that is subsequently not used in the manuscript so could be omitted or moved to supplementary.
- ADAM optimizer has two references [6] and [7] to the same paper.
- In Section 3.3 it is stated that "also included features that are deemed irrelevant to clinicians". Could you please provide some information about these features and how you came to the conclusion that they were not relevant for the task?

**Justification:**

The paper is well-written and easy to follow. The quality of the work could be improved by clarifying the points that I raised above, but overall I would consider this paper above the acceptance threshold.

**Reproducibility:**

Sufficient amount of details available for reproducing the main results, and open access is provided (or promised upon acceptance) to source code and/or data

**Strengths:**

- The paper is well-written and easy to follow.
- Using datasets from multiple demographic regions promotes inclusivity and equity in ML for healthcare.
- Methods that combine high accuracy and allow for interpretability have more potential clinical utility than black-box approaches.

**Summary:**

The paper proposes a few-shot learning framework for Multiple Sclerosis detection. The interpretability of the approach is achieved using DeepSHAP. The method has been trained and evaluated on data from diverse populations with promising results.

**Weaknesses:**

- What was the motivation of approaching the problem with few-shot learning?
- What makes the method inherently interpretable? DeepSHAP could be combined with any other approach for explainability. How is your method more interpretable than other approaches that have used GradCAM to interpret their decisions?
- Why did you select DeepSHAP for interpretability as opposed to other visualization techniques?
- Could you please clarify whether the data splitting was done on a patient-level and that the scans from the same subject were not spread across splits? That would lead to data leakage and inflate the results presented in the paper.
- The evaluation would be more complete if it included some baselines that have been previously proposed for the same task in Table 3.
- What is the intuition behind the result that the model performs better when evaluated on Africal and European datasets? One would expect that the results for an out-of-distribution test set would be worse, therefore this finding is counterintuitive.
- I would suggest in Fig. 2 showing one sample from an MS-positive example and the non-MS sample that is now presented. It would be interesting to compare how the identified features change across diseases.

---

> ### Author Response · Authors · 2024-07-14
> **Rebuttal by Authors**
>
> - We thank the reviewer for the feedback concerning our motivation of using few-shot learning in MS detection. The problem of limited data availability of MS scans especially in the context of African population was the main driver for our motivation to investigate whether FSL would solve the problem of MS detection based on limited data and generalize across datasets. We updated the paper to mention that in the introduction: "Additionally, the collection of MS and other demyelinating disease data is challenging due to the variability in disease presentation, limited patient availability, and the high cost of medical imaging. Therefore, the application of few-shot learning is essential to leverage limited data effectively."
> - We thank the reviewer for their question about GradCAM. Unlike GradCAM, which focuses on visualizing gradient-based attention maps often restricted to convolutional neural networks, DeepSHAP can be applied more broadly across different model architectures and provides a unified measure of feature importance approach to interpretability.
> - We selected DeepSHAP for its unification measure of feature importance. According to their paper (https://arxiv.org/pdf/1705.07874), Deep SHAP is more consistent to human explanation than other methods, as stated in their paper: "We found a much stronger agreement between human explanations and SHAP than with other method"
> - We thank the reviewer for their feedback concerning data leakage prevention. We are sorry for the ambiguity. We rerun the experiment using correct random and patient-wise split to make sure the absence of leakage. We have updated the paper with the newest results.
> - We thank the reviewer the feedback concerning running benchmark experiments. Unfortunately, due to time and resource shortage, this was not possible. We mentioned that in the updated limitation and future studies subsection.
> - We thank the reviewer for the observation. We do not share the reviewer point of view. We think that this observation is rather interesting and requires further investigation in a longer format manuscript. One could initially observe that our model is able to generalize well to unseen distribution.
> - We thank the reviewer for the feedback, we have modified figure 2 accordingly.
> - We thank the reviewer for their feedback concerning the references for the mentioned studies. We apologize for the oversight and have updated the paper accordingly.
> - We thank the reviewer for the feedback. We have named these condition in the introduction.
> - We thank the reviewer for the feedback concerning the missing legend. We apologize for the confusion. We updated Figure 1 missing legend.
> - We thank the reviewer for your feedback concerning the notation table. We apologize for the shortcoming. We used the notations across the manuscript. We think it is essential to group them in a table as a reference for the reader
> - We thank the reviewer for your feedback concerning ADAM optimizer typo reference. We apologize for the oversight. We updated the paper reference.
> - We thank the reviewer for the feedback about the used features for diagnosis, we have updated our manuscript with more details. "We evaluated the explainer results using the key diagnostic features outlined in the McDonald criteria [13], which include lesion size, number of lesions, lesion location, lesion contrast, and lesion shape"
>
> 13: Thompson, A.J et al. : Diagnosis of multiple sclerosis: 2017 revisions of the Mc-Donald criteria 17(2), 162–173. https://doi.org/10.1016/S1474-4422(17)30470-2,

---

### Official Review · Reviewer_BGcm · 2024-07-09

**Recommendation:** 3
**Confidence:** 4

**Clarity:**

The paper is generally clear but has some clarity issues that could be addressed with moderate revision

**Feedback:**

- Please provide more implementation details in Sec. 2.3 under "Prototypical networks".

- In Fig. 2, the authors should provide more example images, and from both classes.

- [page 2] Please define what "RON" is. It is not mentioned anywhere in the paper.

- [Fig. 1] The authors can skip visualizing subfigure (c) since the architecture of the 3D DenseNet-121 is not their contribution, and the current visualization takes up a lot of space that can maybe used better (e.g., more qualitative results for Deep SHAP).

**Justification:**

Although there are some shortcomings (discussed above), I believe they can be addressed with a revision.

**Reproducibility:**

Sufficient amount of details available for reproducing the main results, and open access is provided (or promised upon acceptance) to source code and/or data

**Strengths:**

+ The proposed method is well-described with sufficient details for reproducibility.

+ Utilizing multi-centre datasets, especially from diverse demographics, is helpful for evaluating the robustness of the approach.

**Summary:**

The authors propose DemyeliNeXt, a method for multiple sclerosis (MS) diagnosis from MR images. DemyeliNeXt uses a prototypical network for few-shot learning and the authors use Deep SHAP for explaining its predictions. The proposed method is evaluated on 2 public and 1 private dataset.

**Weaknesses:**

- [page 3] It is unclear what a "task" is.

- In Sec. 3.3, the authors write: "Across all experiments, Experiment E, which involved training on an African dataset and testing on a combination of African and European datasets, achieved the highest classification accuracy.". It is important to note that changing the testing set should not lead to any conclusions about the model performance, since the quantitative results of two experiments that are tested on different sets of images cannot be compared.

- On a related note, on page 6, experiments A through E are not exactly different experiments. D and E just take the trained model from A and test them on different sets of images.

- In Table 3, accuracy is not a good choice of metric since there is a class imbalance in CHU Y.

- [page 1] "However, the model’s lack of explainability raises concerns about the potential misclassification of MS scans due to reliance on spurious correlations." -> This is wrong/misleading. Wang et al.'s model did not have a "lack of explainability", since any post-hoc method can be used for explaining their predictions provided their trained model is available, much like the authors have done with their trained CNN model. The authors should rewrite this sentence.

- [Fig. 2] The Deep SHAP element of the paper seems to be rather hastily discussed without any depth. The authors only provide qualitative results, and only for 1 image. Even then, the results are inconclusive and therefore the entire Deep SHAP section does not appear to contribute anything to their method.

---

> ### Author Response · Authors · 2024-07-14
> **Rebuttal by Authors**
>
> - We thank the reviewer for the feedback concerning the task definition. We apologize for the ambiguity. A task is an few-shot episode which refers to a single learning task or instance that is used to train or evaluate a model. The term task and episode are used interchangeably in the manuscript. We updated our manuscript to have a single notation.
> - We thank the reviewer for the feedback concerning the experiment E. In fact, we believe that having good results on dataset that have never been seen by the trained model shows the ability of model in generalization.
> - We thank the reviewer for the feedback concerning experiment naming. We are sorry for the ambiguity. We have updated the paper. We mentioned: "– Test 1: We used the saved model from Experiment A...Test 2: We used the saved model from Experiment A..."
> - We thank the reviewer for the feedback concerning Table 3. We apologize for the lack of clarity. In order to address the imbalance in CHU Y dataset, we used NON-MS data from CHU X. The number of scans in NON-MS from CHU X test set is 13 scans and the number of MS scans from CHU Y is 91 scans. We have updated our manuscript as follows: "We used the saved model from Experiment A to test on 91 scans from CHU Y MS dataset and on 13 scans CHU X NON-MS test set"
> - We thank the reviewer for the feedback concerning the writing of Wang et al. limitation. We are sorry for the ambiguity. We have rewrited the sentence to be: "However, the model’s explainability is not explored."
> - We thank the reviewer for their feedback concerning Deep SHAP contribution. We are sorry for the ambiguity. In fact, Deep SHAP has detected lesions that are responsible for MS and NON-MS diseases. We updated figure 2 as well as the discussion to be deeper. "The Deep SHAP explainer seems to identify some of the key features for classification, specially the lesions in MS example (Fig.2 B). However, one should note that there is a risk that the included features in the explanation could be deemed irrelevant to clinicians."
> - We thank the reviewer for the feedback concerning ProtoNet implementation details We apologize for the inconvenience. We updated the paper to show the used distance between the support set and the query set. "We employed Euclidian distance for our ProtoNet to calculate the distance between the support samples and query sample"
> - We thank the reviewer for the feedback concerning fig.2. We have updated fig.2 with more examples.
> - We thank the reviewer for their feedback concerning typo RON We apologize for the inconvenience. We updated the paper to state the correct diseases.
> - We thank the reviewer for their feedback concerning fig.1. We have updated the figure accordingly.

---

### Meta-Review · Area_Chair_bjzk · 2024-07-15

**Recommendation:** Accept (Oral)
**Confidence:** 4

**Metareview:**

The authors thoroughly addressed all reviewers' comments and promised to revise the manuscript accordingly when appropriate. Overall, the paper presents a very interesting approach. However, the decision is conditional on all the minor revisions mentioned to the reviewers are in the camera-ready copy, so that page limit would not be violated.

---

### Decision · Program_Chairs · 2024-07-16

**Decision:**

Accept (Oral)

**Comment:**

The paper is accepted with the condition that all minor revisions from the reviewers are included in the final version without exceeding the page limit.